# A Spatio-Temporal Fusion Framework of UAV and Satellite Imagery for Winter Wheat Growth Monitoring

Yan Li [1,2], Wen Yan [3], Sai An [2], Wanlin Gao [1], Jingdun Jia [1], Sha Tao [1,*] and Wei Wang [4,*]

1   College of Information and Electrical Engineering, China Agricultural University, Beijing 100083, China
2   College of Urban Construction, Hebei Normal University of Science and Technology, Qinhuangdao 066004, China
3   Research Center of Marine Economy and Coastal Economic Zone, Hebei Normal University of Science and Technology, Qinhuangdao 066004, China
4   Shijiazhuang Academy of Agricultural and Forestry Sciences, Shijiazhuang 050041, China
*   Correspondence: taos@cau.edu.cn (S.T.); weiwang7109@sina.com (W.W.)

**Abstract:** Accurate and continuous monitoring of crop growth is vital for the development of precision agriculture. Unmanned aerial vehicle (UAV) and satellite platforms have considerable complementarity in high spatial resolution (centimeter-scale) and fixed revisit cycle. It is meaningful to optimize the cross-platform synergy for agricultural applications. Considering the characteristics of UAV and satellite platforms, a spatio-temporal fusion (STF) framework of UAV and satellite imagery is developed. It includes registration, radiometric normalization, preliminary fusion, and reflectance reconstruction. The proposed STF framework significantly improves the fusion accuracy with both better quantitative metrics and visualized results compared with four existing STF methods with different fusion strategies. Especially for the prediction of object boundary and spatial texture, the absolute values of Robert's edge (EDGE) and local binary pattern (LBP) decreased by a maximum of more than 0.25 and 0.10, respectively, compared with the spatial and temporal adaptive reflectance fusion model (STARFM). Moreover, the STF framework enhances the temporal resolution to daily, although the satellite imagery is discontinuous. Further, its application potential for winter wheat growth monitoring is explored. The daily synthetic imagery with UAV spatial resolution describes the seasonal dynamics of winter wheat well. The synthetic Normalized Difference Vegetation Index (NDVI) and Enhanced Vegetation Index 2 (EVI2) are consistent with the observations. However, the error in NDVI and EVI2 at boundary changes is relatively large, which needs further exploration. This research provides an STF framework to generate very dense and high-spatial-resolution remote sensing data at a low cost. It not only contributes to precision agriculture applications, but also is valuable for land-surface dynamic monitoring.

**Keywords:** spatio-temporal fusion (STF); unmanned aerial vehicle (UAV); Sentinel-2 multispectral imager (S2-MSI); growth monitoring





## 1. Introduction

Continuous monitoring of seasonal crop dynamics is critical to coping with climate risk and ensuring food security. Remote sensing imagery provides vital data sources for natural disaster monitoring, crop growth monitoring, yield estimation, and other applications [1–4]. With the development of satellite sensor hardware technology, more and more high-spatial-resolution satellites have been developed to provide more accurate land-surface information. For example, the Sentinel-2 Multispectral Imager (S2-MSI) sensor provides free imagery with a spatial resolution of 10–60 m and a temporal resolution of 5 days by combining Sentinel-2A and Sentinel-2B. Although satellite imagery is inevitably affected by weather conditions, multiple S2-MSI images can be obtained in a crop growing season, which plays an important role in field-scale agricultural applications [5,6]. However, for precision agricultural development, the spatial resolution of S2-MSI imagery is still too coarse to provide more detailed spatial information.

Unmanned Aerial Vehicle (UAV) technology is developing rapidly and has been successfully applied in many fields, including crop dynamic monitoring and yield estimation due to its centimeter-scale spatial resolution and flexibility [7–10]. However, UAV platforms are not fixed like satellites, and each flight test requires manual operation. The seasonal dynamic monitoring of crop growth usually requires continuous imagery, which is costly for UAV observations. Therefore, if the combined advantages of the fixed revisit cycle of the satellite platform and the high spatial resolution and flexibility of the UAV platform can be exploited, it will have great potential for high-precision and long-term continuous monitoring of crop dynamics.

In recent years, researchers have gradually explored the collaborative application of UAV and satellite platforms. Usually, UAV sensors, due to their high-spatial-resolution observations, are applied to obtain precise target information as "truth value", while satellites are used for large-scale and continuous spatial observations. UAV has successfully bridged the scaling gap between ground and satellite data for qualitative or quantitative applications with improved accuracy. A qualitative application may use the land-cover classes identified from UAV imagery to improve the satellite-based classification maps [11–13]. Quantitative applications include taking the information extracted from UAV imagery (plant height, texture, etc.) to assist satellite observation for parameter inversion, or using the quantitative estimations of UAV imagery to adjust those of satellite imagery, thus providing better estimation accuracy [7,14]. In addition, there are also research efforts focusing on the spatio-spectral fusion of UAV and satellite imagery to enrich the spectral range of UAV imagery for land-cover classification [15], or just compare the qualitative or quantitative results from different platforms to investigate their advantages and disadvantages for particular applications or purposes [16,17]. In these studies, the spatial resolution of the final land-cover classification maps or parameter estimations still depends on satellite imagery. In terms of temporal frequency, the remote sensing data of only one or several key growth stages are considered, and the time-series continuous monitoring has not been satisfied.

The current research has confirmed the contribution of the integration of UAV and satellite data to improving application accuracy. Still, their advantages in spatial and temporal resolution have not been fully exploited. Spatio-temporal fusion (STF) takes into account the spatial resolution and temporal resolution of multi-sensors. Hundreds of STF methods have been developed for harmonizing satellite imagery with high temporal resolution but low spatial resolution (coarse imagery) and satellite imagery with high spatial resolution but low temporal resolution (fine imagery). The existing STF methods can be categorized as weighting-based, unmixing-based, learning-based, and hybrid [18]. The spatial and temporal adaptive reflectance fusion model (STARFM) and its enhanced version (ESTARFM) are typical weighting-based methods [19,20]. Fit-FC (regression model Fitting, spatial Filtering and residual Compensation) is a recently proposed weighting-based method that is good at predicting strong temporal changes [21]. Unmixing-based methods, such as the multi-sensor multiresolution technique (MMT) and Unmixing-Based Data Fusion (UBDF), unmix coarse pixels into high spatial resolution based on the linear spectral mixing theory [22,23]. Learning-based methods use machine learning algorithms to learn the relationships between coarse and fine imagery pairs and then predict unobserved fine imagery [24]. Hybrid methods such as Flexible Spatiotemporal DAta Fusion (FSDAF) integrate procedures from the above two or three categories [25]. In addition, optimization frameworks are presented to improve the fusion effect of existing methods [26,27].

Each developed STF approach shows applicability and limitations for a particular situation, and many studies have compared the typical methods [28–30]. However, how the existing STF methods perform in UAV and satellite imagery fusion remains unknown. Furthermore, how to combine the spatial and temporal advantages of UAV and satellite platforms for agricultural applications remains unexplored.

This research aims to fill this gap by establishing an STF framework of UAV and satellite imagery for continuous winter wheat growth monitoring. Our overarching goal is

to exploit the combined advantages of the fixed revisit cycle of the satellite platform and the high spatial resolution and flexibility of the UAV platform, so as to generate continuous imagery with high spatial and temporal resolution for precision agricultural monitoring.

The paper is organized around three components. Firstly, the adaptability of four state-of-the-art STF methods (STARFM, Fit-FC, UBDF, and FSDAF) to the fusion of UAV and satellite imagery is investigated. Further, an STF framework of UAV and satellite imagery is constructed according to the characteristics of UAV and satellite platforms, and daily synthetic datasets with the spatial resolution of UAV imagery are generated. Finally, the application potential of synthetic imagery for continuous crop growth monitoring is explored.

## 2. Materials and Preprocessing

### 2.1. Study Site

The study site is located in Shijiazhuang Academy of Agriculture and Forestry Sciences (38°6′ N, 114°31′ E) in Hebei province, China (Figure 1). Its annual average temperature and rainfall are 14.3 °C and 500 mm, respectively. The total area of the experimental field is approximately 120 m × 120 m. The winter wheat was sown on 8 October 2021 and harvested on 15 June 2022, with a growth period of about 240 days. In particular, the green-up stage occurred from about late February to early March, the jointing stage lasted from about mid-March to mid-April, the heading-filling stage ran from about late April to early May, and the milk stage began from about mid-May to late May.

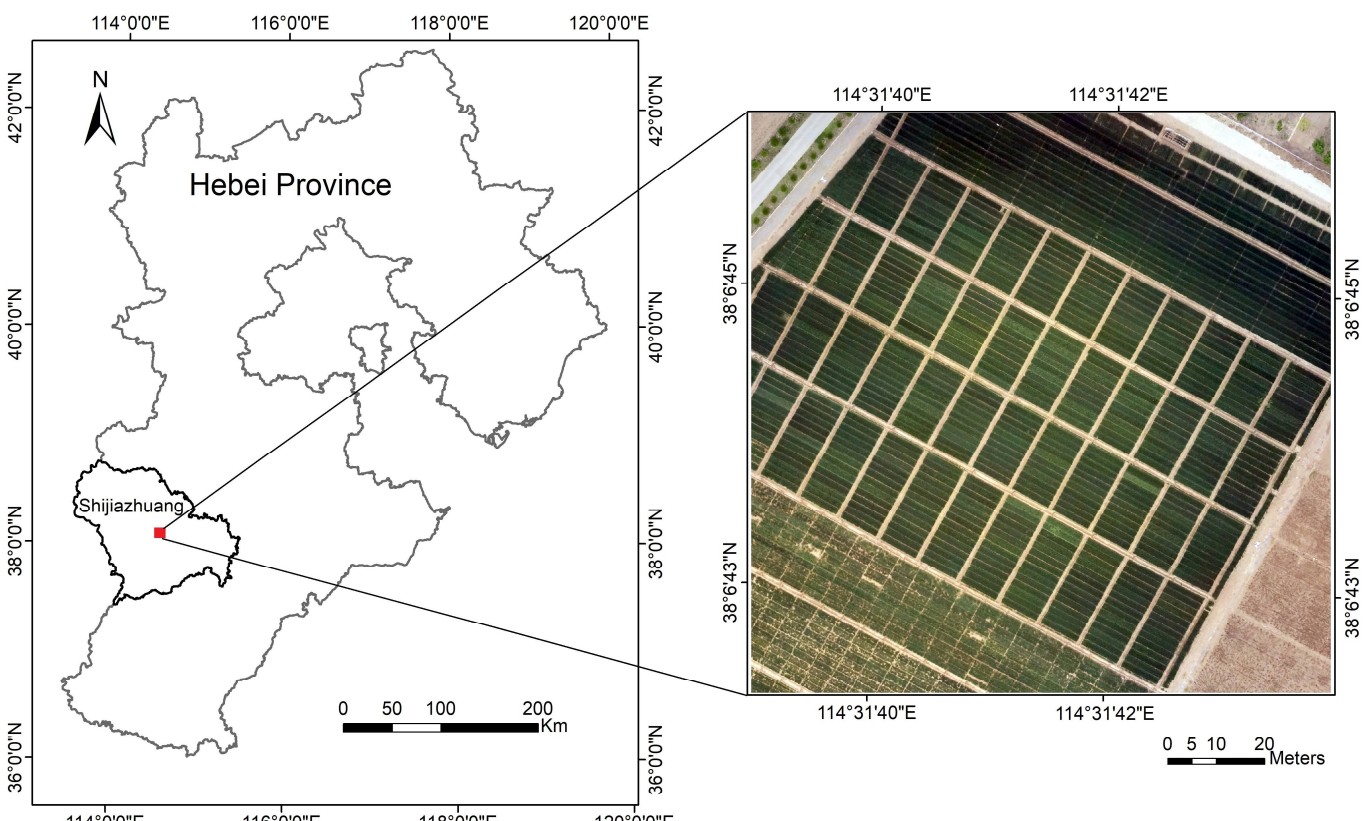

**Figure 1.** Location of the study site (The color map is a Red-Green-Blue band composite of UAV imagery on 12 April 2022). Reference system is WGS 1984-UTM 50 N. Representation scales are 1:3,350,000 (**left**) and 1:700 (**right**).

### 2.2. Data Sources and Preprocessing

This experiment was conducted from March to June 2022, which covered the main growth stages of winter wheat. The data sources used in this research contain S2-MSI

imagery and UAV multispectral imagery. The S2-MSI Level-1C products were freely downloaded from https://scihub.copernicus.eu/ accessed on 30 June 2022. There was nine cloud-free S2-MSI images covering this study area on DOY 67, DOY 87, DOY 92, DOY 102, DOY 122, DOY 137, DOY 147, and DOY 152 (Figure 2). The atmospheric correction was implemented using the Sen2Cor processor (version 2.8). Manual geometric correction was completed with ground control points (GCPs) in ENVI software. To correspond to the spectral bands of the UAV data, only the blue, green, red, red-edge (Red-E), and narrow near-infrared (NIR) bands were considered in STF. To enhance the spatial resolution of S2-MSI imagery, the Red-E and NIR bands of 20 m were reconstructed using the 10 m bands to produce 10 m S2-MSI imagery using the SupReME algorithm [31].

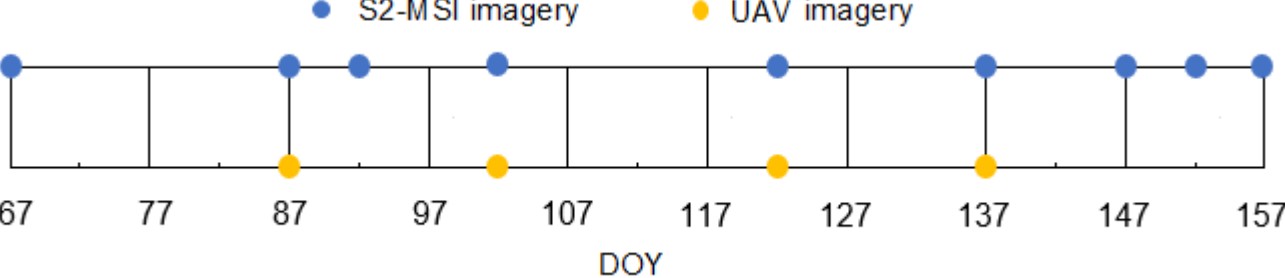

**Figure 2.** DOYs of UAV and S2-MSI imagery available during the monitoring period.

The UAV imagery was collected from a four-rotor consumer UAV equipped with an integrated multispectral imaging system (DJI Phantom 4 multispectral). It incorporates an RGB camera and five multispectral cameras (receiving blue, green, red, Red-E, and NIR bands) for visible and multispectral imaging, respectively (Table 1). The platform is equipped with real-time kinematic (RTK) GNSS and flies autonomously using pre-planned flight missions created with DJI ground station Pro software. A total of four UAV flights were carried out in the early jointing stage (DOY 87), late jointing stage (DOY 102), heading-filling stage (DOY 122), and milk stage (DOY 137), respectively. All missions held a planned flying altitude of 100 m and a constant speed of 5 m/s, resulting in a ground sampling distance (GSD) of approximately 5 cm/pixel. Successive imagery along the flight lines was acquired with a forward and side overlap of 80% between 11:00 a.m. and 2:00 p.m. in clear-sky conditions. The UAV specification and parameters are detailed in Table 2. Prior to the UAV flights, 12 GCPs were evenly placed in the field using black and white tiles with a length and width of 1 m, and the location information was obtained with a GPS system (S-GNSS, China). Also, two standard reflective cloths were laid out for radiometric correction. The UAV system and ground targets are shown in Figure 3.

**Table 1.** Band information of UAV and S2-MSI imagery used for STF.

| | **S2-MSI** | | | **UAV** | | |
| Band Name | ID | Central Wavelength (nm) | Spatial Resolution (m) | ID | Central Wavelength (nm) | Spatial Resolution |
| --- | --- | --- | --- | --- | --- | --- |
| Blue | 2 | 490 | 10 | 1 | 450 | H/18.9 |
| Green | 3 | 560 | 10 | 2 | 560 | H/18.9 |
| Red | 4 | 665 | 10 | 3 | 650 | H/18.9 |
| Red-E | 6 | 740 | 20 | 4 | 730 | H/18.9 |
| NIR | 8a | 865 | 20 | 5 | 840 | H/18.9 |

**Table 2.** UAV specification and parameters.

| Parameter Name | Parameter Specification |
| --- | --- |
| focal length | 5.74 mm |
| camera angle | $-90°$ |
| altitude | 100 m |
| speed | 5 m/s |
| GSD | 5 cm/pixel |
| forward overlap | 80% |
| side overlap | 80% |

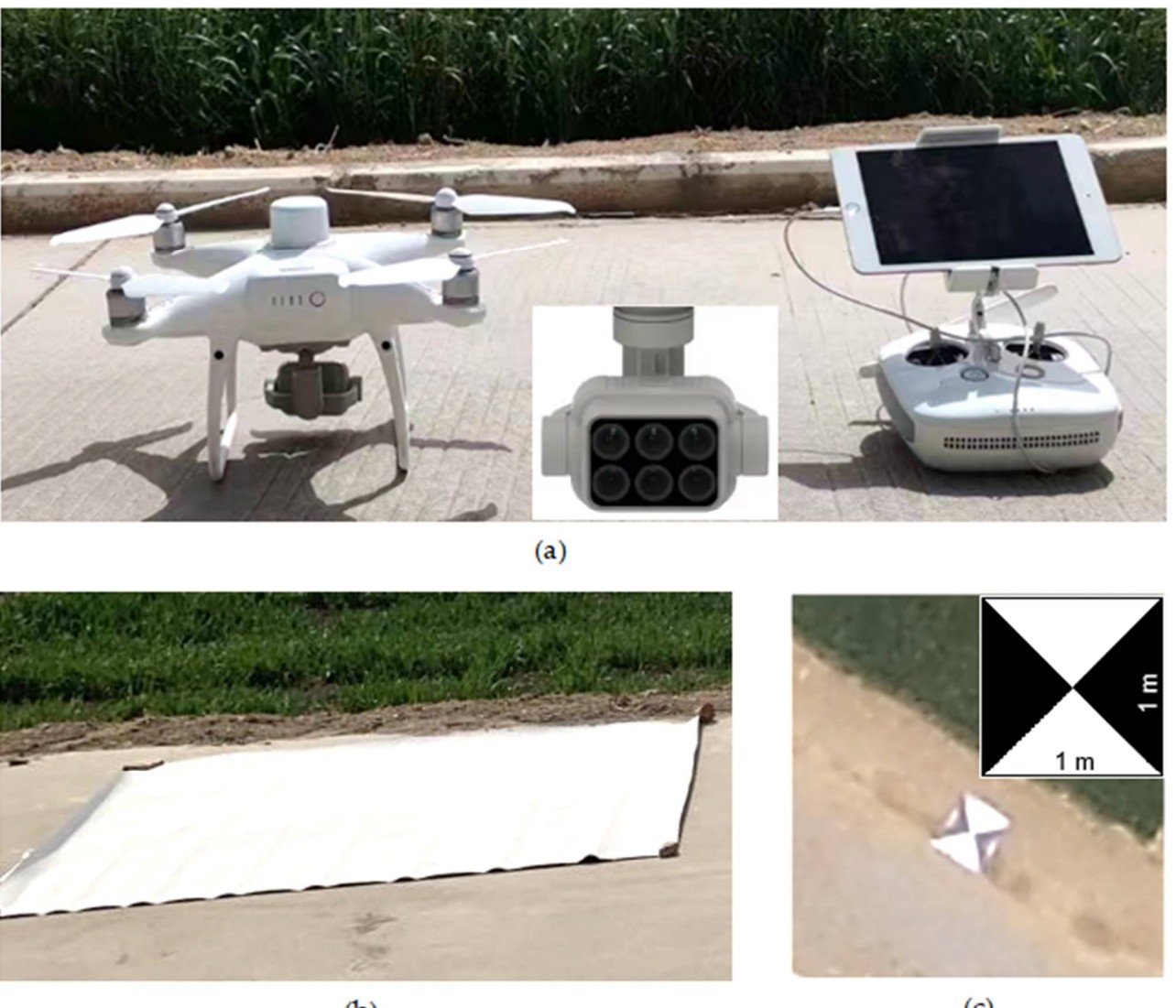

**Figure 3.** UAV system and ground targets: (**a**) UAV with multispectral camera and remote control; (**b**) standard reflective cloth; (**c**) GCP.

UAV imagery preprocessing mainly included image stitching, radiometric correction, and geometric correction. The imagery collected using UAV was first processed using the Agisoft Photoscan Professional software. A photogrammetric workflow with default settings was applied, in turn, for photo addition and alignment, dense point cloud building, mesh and texture generation, and digital surface model (DSM) and orthomosaic production. Thus, the orthophoto map with 0.05 m spatial resolution and WGS-84 coordinate system for each band was obtained.

In ENVI software, the orthophoto maps of multiple bands were relatively registered and synthesized to a single image with multiple spectral bands. The radiometric calibration was performed to convert the image digital numbers (DNs) into surface reflectance using two standard reflective cloths, with reflectivity of 30% and 70%, respectively. The widely used linear empirical correction method was used. It calculates a model based on the relationship between the DNs of the standard reflective cloths and the surface reflectance within an orthomosaic. Then, the UAV data was reprojected to match the S2A-MSI imagery. Although the UAV is equipped with RTK, the UAV imagery was further geo-registered using the 12 GCPs to improve geometric accuracy. The registration error of each band was less than one UAV pixel (0.05 m) and the average RMS error of the 12 GCPs in each band of UAV imagery on DOY 102 is presented in Table 3. To eliminate the random error caused by the reflectance of a single point and to blend with S2-MSI imagery, the UAV data were resampled to 0.5 m using the average values within a $10 \times 10$ pixels window.

**Table 3.** The RMS error in geo-registration of UAV imagery on DOY 102.

| Band Name | RMS Error (Pixel) |
| :---: | :---: |
| Blue | 0.4625 |
| Green | 0.5186 |
| Red | 0.6547 |
| Red-E | 0.5286 |
| NIR | 0.4293 |

## 3. Methods

The STF framework of UAV and satellite imagery developed in this research involves four steps (Figure 4). After the UAV and S2-MSI data were resampled to the required spatial resolution, the UAV data were shifted to register with the S2-MSI imagery to reduce the influence of registration errors. Then, the relative radiometric normalization was performed to mitigate their radiometric differences. After this, the UAV and S2-MSI imagery were blended using existing STF methods for multi-source satellite imagery to generate the preliminary reflectance predictions. Finally, to further improve the fusion results, the Consistent Adjustment of the Climatology to Actual Observations (CACAO) approach was introduced. A phenology model established using the preliminary reflectance predictions was fitted to the UAV observations, so the daily synthetic UAV-LIKE reflectance was generated.

### 3.1. Registration of UAV and Satellite Imagery

The spatial resolution of the S2-MSI imagery is relatively rough compared with the UAV data. Although the S2-MSI imagery has been manually registered with GCPs, there are still registration errors among UAV and S2-MSI imagery, which will greatly affect the fusion accuracy. Especially when the spatial scale of UAV and satellite imagery is large, the influence of registration error is great [32]. To mitigate the registration error, the UAV and S2-MSI imagery were co-registered. First, since the pixel of S2-MSI imagery is rough, and the one-pixel shift will produce a large distance, we shifted the UAV imagery in both east–west and north–south directions by 0.5 m. Considering that the S2-MSI imagery has been manually registered, the maximum shift in each direction was set to 2 m (4 UAV pixels), with a total of 46 shift positions. Then, the UAV data were downsampled to the same spatial resolution as the S2-MSI imagery using the pixel aggregation method. The resulting root mean square error (RMSE) values between the corresponding spectral bands of the downsampled UAV and S2-MSI imagery were then computed. The co-registration was optimized by minimizing the RMSE between all spectral bands of each image pair. Finally, the co-registered UAV and satellite images were cropped to cover the same area.

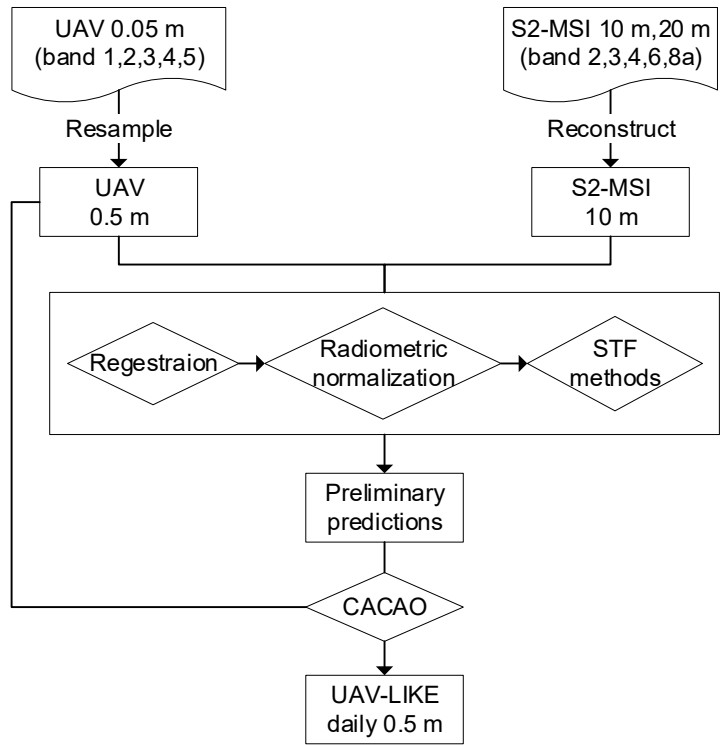

**Figure 4.** Flowchart of the STF framework for UAV and satellite imagery.

### 3.2. Relative Radiometric Normalization

Due to the different spectral response functions of multi-sensors, radiometric inconsistency is another major factor affecting the fusion accuracy, especially for UAVs and satellites with very different sensors, platform heights, and imaging conditions [33]. Therefore, it is necessary to minimize the radiation differences among multi-source imagery. We downsampled the UAV imagery to 10 m, comparable to S2-MSI, and established linear regression models for relative radiometric normalization.

The linear regression model between the downsampled UAV imagery ($UAV_{t_0}^{down}$) and S2-MSI imagery ($S_{t_0}$) at base time $t_0$ was established. A moving window was used to fit the regression model for each center S2-MSI pixel of each band.

$$UAV_{t_0}^{down}(x,y) = \frac{1}{r^2}\sum_{i=1}^{r^2} UAV_{t_0}(x_i,y_i) = a_0 \times S_{t_0}(x,y) + b_0 + \varepsilon_0, \tag{1}$$

where, $a_0$ and $b_0$ are the regression coefficients, $r$ is the spatial scale of UAV and S2-MSI imagery, and $\varepsilon_0$ is the error. There are $n_w$ pixels within the local window $w \times w$. The fitting coefficients ($\hat{a}_{t_0}^w, \hat{b}_{t_0}^w$) in the window were obtained using the least square method:

$$(\hat{a}_{t_0}^w, \hat{b}_{t_0}^w) = \arg min \sum_{i=1}^{n_w} ||a_{t_0}^w \times S_{t_0}^i(x,y) + b_{t_0}^w - UAV_{t_0}^{down_i}(x,y)||, \tag{2}$$

The S2-MSI imagery at $t_0$ was then corrected by the coefficients in the corresponding window as the normalized imagery ($\hat{S}_{t_0}$):

$$\hat{S}_{t_0}(x,y) = \hat{a}_0 \times S_{t_0}(x,y) + \hat{b}_0, \tag{3}$$

The similar S2-MSI imagery ($S_{t_p}^s$) at predicted time $t_p$ was calculated from the increment of S2-MSI imagery and used to build the linear relationship with S2-MSI imagery at $t_p$ ($S_{t_p}$).

$$S_{t_p}^s(x,y) = UAV_{t_0}^{down}(x,y) + S_{t_p}(x,y) - S_{t_0}(x,y)$$
$$= a_p \times S_{t_p}(x,y) + b_p + \varepsilon_p, \tag{4}$$

where, $a_p$ and $b_p$ are regression coefficients and $\varepsilon_p$ is the error. The fitting coefficients $(\hat{a}_{t_p}^w, \hat{b}_{t_p}^w)$ in the window can be calculated using the least square method.

Then, the normalized imagery at $t_p$ $(\hat{S}_{t_p})$ was calculated as follows:

$$\hat{S}_{t_p}(x, y) = \hat{a}_p \times S_{t_p}(x, y) + \hat{b}_p, \tag{5}$$

A moving window with a radius of 2 S2-MSI pixels was set in this study. After relative radiometric normalization, the reflectance of UAV and satellite imagery was more comparable and consistent, which is beneficial to the posterior STF process.

### 3.3. Preliminary Predictions Using Existing STF Methods

Based on the registered and radiometrically normalized UAV and S2-MSI imagery, four typical STF methods, including STARFM, Fit-FC, UBDF, and FSDAF, were applied to generate the preliminary temporal reflectance predictions. The learning-based methods were not considered, because the pixels of S2-MSI imagery corresponding to the coverage area of UAV imagery were not enough for model training [34].

There were four UAV and S2-MSI imagery pairs sharing the same date, which were used as the base imagery pair. The time-series synthetic imagery was predicted from the adjacent base imagery in front of the predicted dates. In particular, DOY 92 and DOY 102 were predicted from DOY 87, DOY 122 was predicted from DOY 102, DOY 137 was predicted from DOY 122, and DOY 147, DOY 152, and DOY 157 were predicted from DOY 137. Since no UAV data were available before DOY 67 and DOY 87, the synthetic imagery for these two dates was backward-predicted from the base imagery on DOY 87 and DOY 102, respectively. Finally, a sequence of UAV-LIKE imagery with the same dates as S2-MSI imagery was generated.

Referring to relevant studies and our tests, the parameters of the STF methods were carefully selected [26,30]. For STARFM, the radius of the moving window was set to 15 UAV pixels, the number of categories was set to 5, and the k-means method was used for classification. For Fit-FC, the window size was set to $5 \times 5$ S2-MSI pixels in regression model fitting (RF) and $31 \times 31$ UAV pixels in spatial filtering (SF) and residual compensation (RC), and the number of similar pixels within each local window was 20. For UBDF, the window size was set to $9 \times 9$ S2-MSI pixels, and the classification map of 5 categories was used. For FSDAF, the window size of $31 \times 31$ UAV pixels, the similar pixels of 20, and the classification map of 5 categories were used.

### 3.4. Reflectance Reconstruction with CACAO

Due to its successful application in STF [35,36], CACAO, a gap-filling and temporal smoothing method [37], was introduced here for high spatiotemporal reflectance reconstruction. CACAO minimizes the bias between seasonal phenology models and UAV observations with temporal constraints. It consists of two steps:

First, the preliminary temporal reflectance predicted with the existing STF methods was smoothed using the Savitzky–Golay (SG) filtering method [38] to construct the seasonal phenology models for each pixel of each band. SG smoothed the reflectance sequence by fitting a low-degree polynomial within a temporal window. Referring to our tests, a 2-degree polynomial and a temporal window with a width of 17 days were set.

Then, the seasonal phenology model of each pixel was fitted to actual UAV observations one by one. It was scaled and shifted to minimize the RMSE with the UAV data:

$$arg \min_{scale, shift} \sqrt{\frac{1}{n} \sum_{i=1}^{n} \left( R_{UAV}(t_i) - scale \times R_{pm}(t_i + shift) \right)^2}, \tag{6}$$

where $R_{UAV}$ and $R_{pm}$ denote reflectance data of the UAV imagery and phenology model, respectively. $n$ is the available number of UAV images (here, $n = 4$), and $t_i$ denotes the time in DOYs. *scale* and *shift* are the parameters used for phenological model adjustment.

The synthetic UAV-LIKE imagery at arbitrary dates can be calculated from the phenology model to generate the continuous daily dataset.

In this study, the four existing STF methods were compared for UAV and satellite imagery fusion. The most suitable method was identified and coupled with CACAO, producing a CA-STF method. Thus, the preliminary reflectance predictions were consistently adjusted to reconstruct more accurate time-series reflectance data with UAV spatial resolution.

### 3.5. Accuracy Assessment of the STF Framework

The assessment of the STF framework contains two aspects: the accuracy assessment of the reflectance predictions and the application potential of the STF framework for time-series growth monitoring. To assess the all-around performances of the STF methods, a novel assessment framework proposed by Zhu et al. was introduced [39]. It quantitatively assesses both spectral and spatial fusion accuracy with less information redundancy. There are four accuracy metrics, including RMSE, average difference (AD), Robert's edge (EDGE), and local binary pattern (LBP), in the assessment framework. The metrics range from −1 to 1 except RMSE, which ranges from 0 to 1. Negative values mean underestimation, positive values indicate overestimation, and values closer to 0 indicate smaller spectral and spatial errors. In addition, we observed the false color composition of the synthetic UAV-LIKE imagery to check the spatial details and displayed the density scatterplots of reflectance for each band with linear fit equations. According to the available dates of UAV and satellite imagery, DOY 102, DOY 122, and DOY 137 were selected as the validation dates. A leave-one-date-out cross validation was conducted, and the verification date was not involved in STF.

Further, to evaluate the application potential of the STF framework for time-series growth monitoring, the vegetation indices (VIs) computed from synthetic UAV-LIKE and actual UAV imagery were compared. The widely used Normalized Difference Vegetation Index (NDVI) [40] and Enhanced Vegetation Index 2 (EVI2) [41] were selected. The VI values within a 2 × 2 pixels window were averaged to alleviate the potential for sampling errors. We analyzed the consistency of seasonal VIs of three plots randomly selected from the study site. Further, the absolute errors of VIs on DOY 102 were visualized to observe the spatial error distribution. The code and software settings in the spatio-temporal fusion framework are presented in Table A1 (Appendix A).

### 4. Results

#### 4.1. Fusion Accuracy Evaluation of Different STF Methods

4.1.1. Quantitative Evaluation

The spectral and spatial metrics (AD, RMSE, EDGE, LBP) between the actual UAV observations and the synthetic UAV-LIKE imagery generated using various STF methods on the three validation dates are shown in Figure 5. Except for UBDF, the other three existing STF methods do not show a significant difference in spectral metrics. However, the spatial predictions of all four existing STF methods are generally poor, especially for the edge features, which are underestimated (EDGE values are negative). Specifically, UBDF produces significantly larger spectral and spatial errors than other methods. Especially for spatial metrics, all LBP values are negative, and the largest error is −0.3861 (NIR band on DOY 137), which is nearly 17 times higher than FSDAF (−0.0228). It indicates that UBDF is not competent in predicting textural features. Although Fit-FC provides better spectral predictions than UBDF, its spatial predictions are also poor, and the EDGE metrics are even worse than UBDF, reaching a maximum of −0.5528 (Red-E band on DOY 122). Compared with UBDF and Fit-FC, STARFM and FSDAF improve the spectral and spatial predictions; the spatial features especially are obviously improved. The quantitative prediction accuracy of the two methods is similar; STARFM performs slightly better in spectral prediction, while FSDAF performs marginally better in spatial prediction. Considering the running

time of the methods, STARFM was regarded as a relatively competent method to fuse UAV and S2-MSI imagery among the four tested existing STF methods in this research.

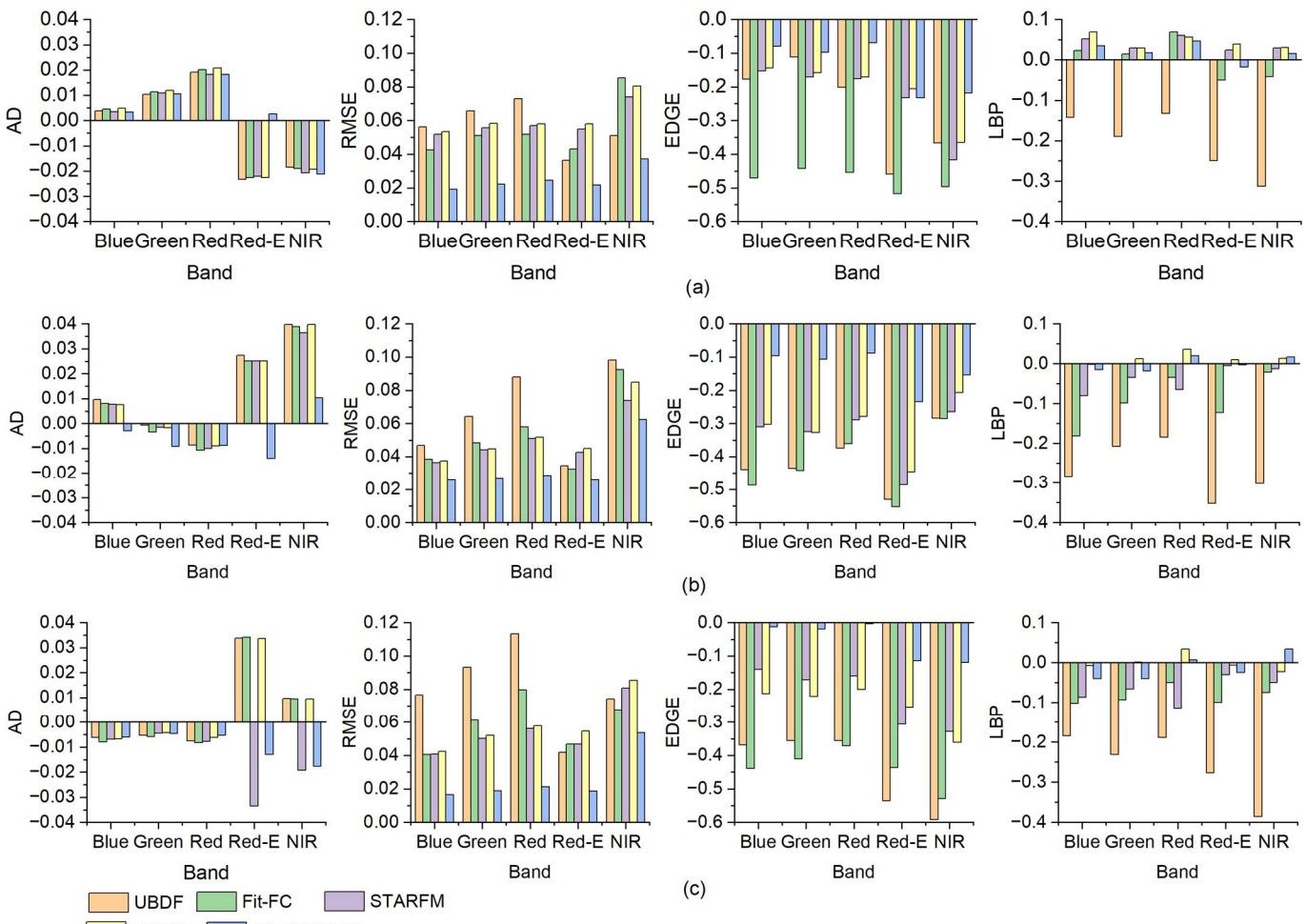

**Figure 5.** Quantitative metrics of different STF methods on (**a**) DOY 102, (**b**) DOY 122, and (**c**) DOY 137.

Combining STARFM with CACAO to generate the CA-STARFM method, the prediction ability of CA-STARFM was investigated. The quantitative metrics are also shown in Figure 5. It is obvious that CA-STARFM presents excellent predictions for both spectral and spatial features. Especially for edge features, CA-STARFM shows significant improvement. Compared with STARFM, the absolute values of AD and RMSE of CA-STARFM decrease by 0.0206 (Red-E band on DOY 137) and 0.0365 (red band on DOY 137) at the maximum, respectively. The absolute values of EDGE and LBP decrease by a maximum of 0.2513 (Red-E band on DOY122) and 0.1065 (red band on DOY 137), respectively. The fusion accuracy for Red-E and NIR bands is slightly worse than that of the blue, green, and red bands, which may be related to the difference in band settings between UAV and S2-MSI sensors (Table 1).

### 4.1.2. Visual Assessments

To investigate the spatial details of the synthetic UAV-LIKE imagery, the NIR-Red-Green band composite on DOY 102 is shown in Figure 6. In general, UBDF and Fit-FC lead to severe blurring effects, while other methods generate synthetic imagery closer to actual observations. Specifically, the UBDF predictions result in large errors in spectral color, spatial boundary, and texture. The Fit-FC predictions are similar to actual observations in spectral color, but the spatial boundary is significantly smoothed, which is also reflected by the negative EDGE metrics (Figure 5). The imagery predicted with STARFM and

FSDAF are barely distinguishable, and the quantitative metrics are close. Comparing CA-STARFM with STARFM and FSDAF, in the top right square box, it is observed that CA-STARFM performs significantly better than STARFM and FSDAF in predicting ground object boundaries. A close inspection of the rectangle in the bottom left corner reveals that CA-STARFM captures the detail features of the object boundaries better. The bottom right ellipse also shows that CA-STARFM is competent in predicting spatial textures. Therefore, CA-STARFM is more capable of capturing the spatial boundary and texture of non-planting areas. For planting areas, the spectral color of CA-STARFM predictions is more accurate, but it is difficult to observe the spatial differences between CA-STARFM, STARFM, and FSDAF visually.

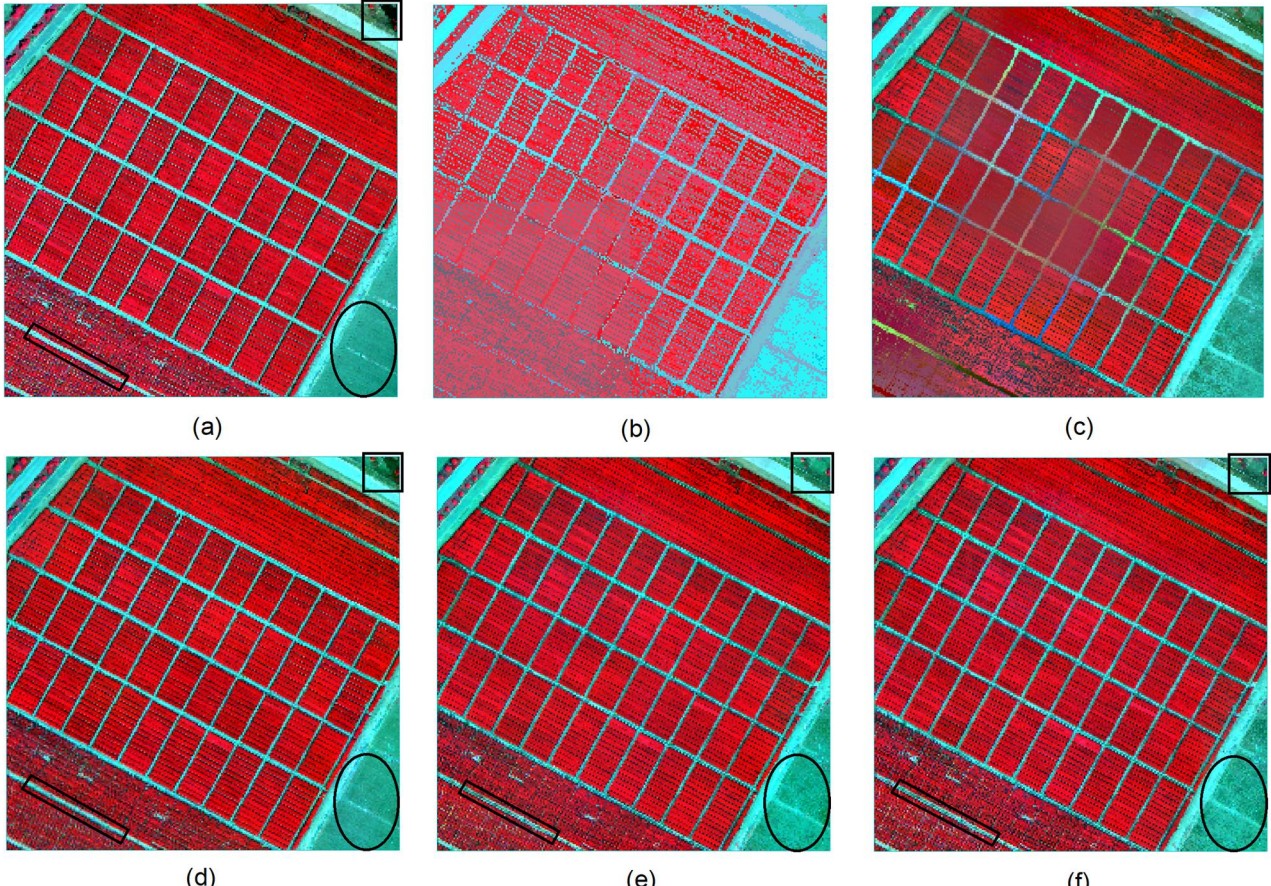

**Figure 6.** NIR-Red-Green band composites of (**a**) UAV observations, (**b**) predictions of UBDF, (**c**) predictions of Fit-FC, (**d**) predictions of CA-STARFM, (**e**) predictions of STARFM, (**f**) predictions of FSDAF. Reference system is WGS 1984-UTM 50 N. Representation scale is 1:700.

4.1.3. Scatter Plots of Reflectance

To further observe the dispersion of the reflectance predictions, taking DOY 102 as an example, the density scatterplots between actual the UAV observations and the predictions provided by STARFM and CA-STARFM are shown in Figure 7. The linear fit equation and correlation coefficient (r) for each band are displayed. It is observed that CA-STRAFM predicts high-density points distributed along the 1:1 line. Better fitting coefficients and higher r values are obtained for each band, especially for the blue, green, and red bands, with r higher than 0.95. A large number of outliers predicted by STARFM are adjusted by CA-STARFM, and the most apparent improvement is observed in the Red-E band, where the slope and r increase by nearly 0.36 and 0.43, respectively. Therefore, CA-STARFM achieves a better match with actual UAV observations than STARFM, which is also reflected in the quantitative metrics (Figure 5). However, the relatively poor performance of the NIR band deserves further exploration.

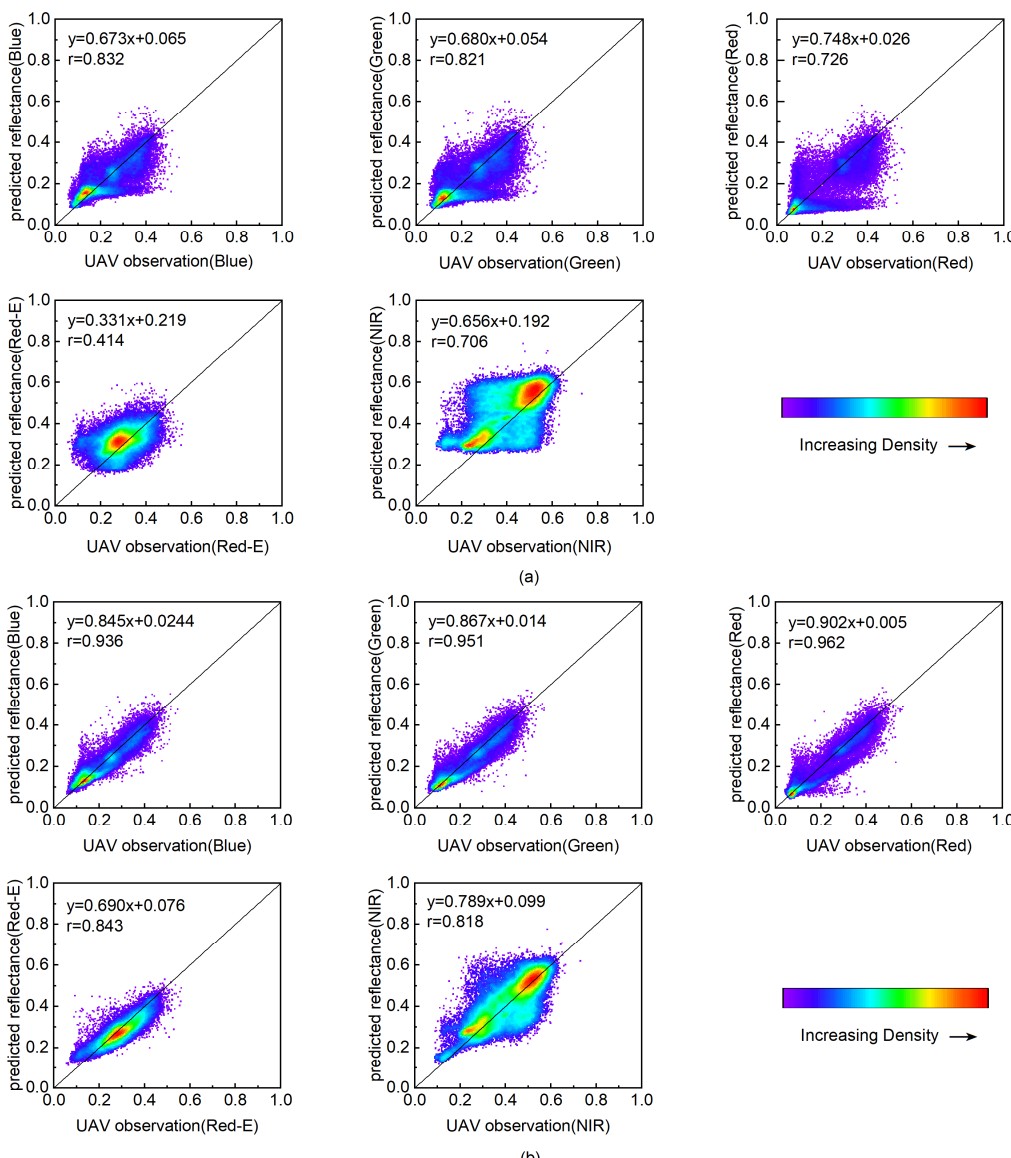

**Figure 7.** Scatterplots of reflectance between actual observations and predictions on DOY 102. (**a**) predictions of STARFM, (**b**) predictions of CA-STARFM.

### 4.2. Accuracy Analysis of VIs Generated from CA-STARFM Blended Imagery

#### 4.2.1. Seasonal Variation in VIs

The introduction of the CACAO method makes it possible to generate imagery at arbitrary dates during the monitoring period, and the dense temporal data are particularly suitable for continuous vegetation monitoring. Three plots were randomly selected to explore the seasonal variation in NDVI and EVI2 generated from the STARFM and CA-STARFM blended imagery, and their differences from the actual UAV observations are shown in Figure 8. CACAO shifted the sequence preliminary predictions to adjust and optimize the predictions. In this study, the shift was set to a maximum of 10 days, so the time-series data generated with CA-STARFM were from DOY 77 to DOY 147. It is observed that both NDVI and EVI2 describe crop growth trends well. From the early jointing stage on DOY 77, leaves, stems, roots, ears, and other organs of winter wheat grow rapidly at the same time, and NDVI and EVI2 show a rapid increasing trend. Around the end of the jointing stage on DOY 114, which is the most vigorous stage, NDVI and EVI2 reach a peak. Then entering the heading-filling stage, winter wheat begins to age gradually, and NDVI and EVI2 show a declining trend. After about DOY 140, wheat ears and leaves turn yellow,

and NDVI and EVI decrease rapidly. In general, the synthetic time-series VIs of SATRFM and CA-STARFM are consistent with the UAV observations, except on DOY 87when they were subject to a slightly larger overestimation. Moreover, the synthetic VIs of CA-STARFM are closer to UAV observations with smaller deviation and continuous temporal resolution. This indicates that CA-STARFM is competent to monitor winter wheat seasonal dynamics more accurately and intensively than the tested existing STF methods.

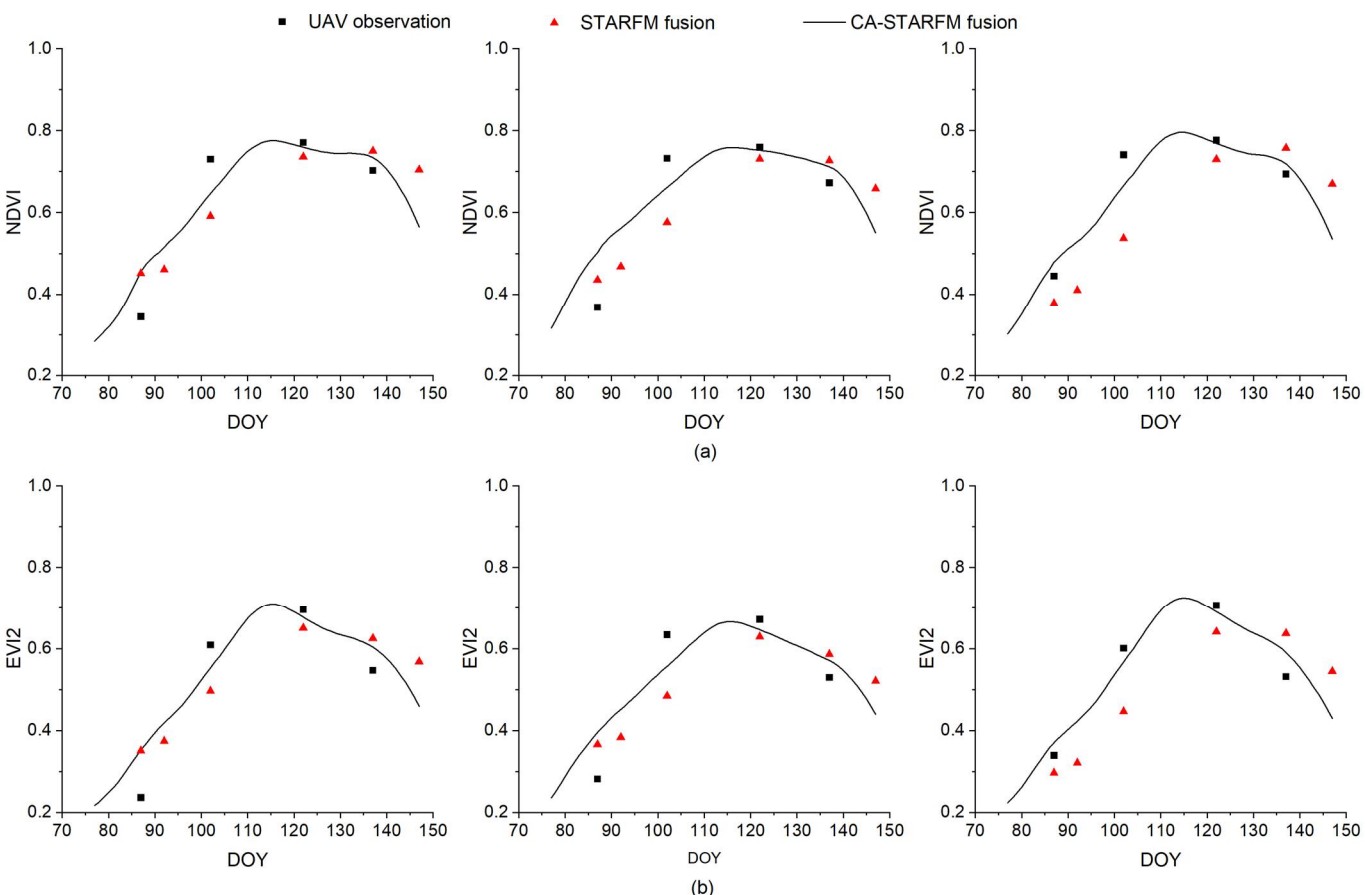

**Figure 8.** Seasonal dynamic monitoring of observed and synthetic VIs for three random plots. (**a**) NDVI, (**b**) EVI2.

4.2.2. Spatial Distribution of Absolute Error of VIs

To further observe the spatial error distribution of the synthetic VIs, Figure 9 visualizes the absolute error of NDVI and EVI2 on DOY 102 and their probability density distribution. In Figure 9, the first row displays the VI error analysis of STARFM, and the second row shows the VI error analysis of CA-STARFM. It is obvious that the VI error of CA-STARFM is significantly reduced. Overall, the VI residual distributions of STARFM and CA-STARFM show peak shapes centered at 0 without significant bias, but those of CA-STARFM are more evenly distributed with a small range of variation. The pixels with large errors are generally the boundaries between winter wheat and bare soil, and most of the pixels are overestimated. The winter wheat pixels show much better predictions. Moreover, EVI2 produces fewer outliers with lower values than NDVI. Therefore, the error analysis confirms that the synthetic VIs of CA-STARFM are in good agreement with the observed values, and EVI2 is more robust than NDVI.

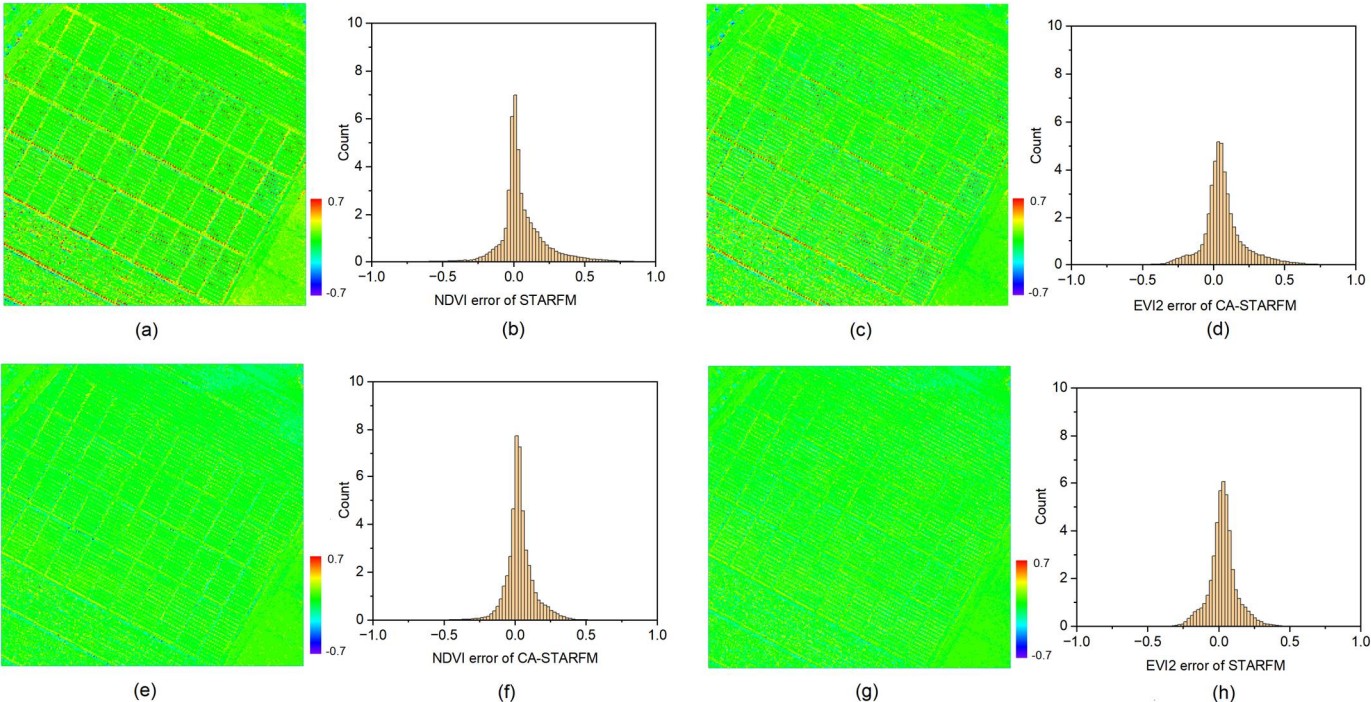

**Figure 9.** Error analysis of VIs. (**a**) spatial distribution of NDVI error of STARFM, (**b**) probability density distribution of NDVI error of STARFM, (**c**) spatial distribution of EVI2 error of STARFM, (**d**) probability density distribution of EVI2 error of STARFM, (**e**) spatial distribution of NDVI error of CA−STARFM, (**f**) probability density distribution of NDVI error of CA−STARFM, (**g**) spatial distribution of EVI2 error of CA−STARFM, (**h**) probability density distribution of EVI2 error of CA−STARFM. Reference system is WGS 1984−UTM 50 N. Representation scale is 1:700.

## 5. Discussion

### 5.1. Analysis of the STF of UAV and Satellite Imagery

The STF of multi-source satellite imagery has been focused on for a long time, and the fusion methods are basically mature. However, the quantitative agricultural application of UAV technology has rapidly expanded only in recent years, and the integration of UAV and satellite is just developing. Rare studies have explored the STF of UAV and satellite imagery. There are considerable differences in spatial resolution, sensor design, and imaging conditions between UAV and satellite platforms, so the satellite-to-satellite STF methods cannot be directly applied to the STF of UAV and satellite imagery. In this study, the S2-MSI imagery was reconstructed at a spatial resolution of 10 m with SupReME to reduce the spatial ratio between UAV and satellite imagery. Since SupReME has been verified to well maintain the spectral and spatial consistency of S2-MSI imagery [31,42], we did not display the verification again. Then a relative radiometric normalization method was applied to mitigate the influence of sensor design. However, the relationship was simply assumed to be linear, and a better radiometric correction method will help improve imagery consistency. We tested four widely used existing STF methods for UAV–satellite fusion, but all of them were found to produce large fusion errors. Therefore, the CACAO method was further introduced to adjust the preliminary predictions, which significantly improved the spectral accuracy and spatial texture features.

CACAO aims to reduce the temporal noise of phenological models to approximate UAV observations. At least two imagery pairs on the base dates are required, and more fine imagery contributes to capturing the temporal and spatial changes. Due to the flexibility of UAV platforms, it is not difficult to obtain more than two UAV images in a crop growing season. Therefore, the STF framework proposed in this research has great potential for UAV and satellite imagery fusion. In addition, the phenological model constructed using

preliminary predictions plays a critical role in the STF framework, and more competent STF methods are worth exploring in subsequent research.

*5.2. Application Potential of Time-Series Growth Monitoring*

Previous studies have exhibited the applicability and great potential of STF of multi-source satellite imagery for continuous crop growth monitoring [43,44]. However, the temporal frequency of synthetic imagery generated using the existing STF methods depends on the available dates of coarse imagery. The demand for growth monitoring with subweekly frequency, which is the international consensus [45], is still hard to meet due to the influence of weather conditions. The particular advantage of the STF framework proposed in this study is that the discontinuous raw data can be fused to generate daily time-series data. As long as the adjustment coefficients are calculated, synthetic imagery at an arbitrary date with high spatial resolution (here, 0.5 m) can be generated quickly. Therefore, the new STF framework is especially suitable for the application of time-series continuous data with high spatial resolution, such as the seasonal dynamic monitoring of crop growth. We verified the application potential of the STF framework for vegetation growth monitoring by considering the temporal variation in VIs and the spatial distribution of their errors. Although the synthetic VIs conform to the observations, there are still underestimates or overestimates. The predictions on a specific date are usually similar to the base imagery. DOY 102, DOY 122, and DOY 137 were forward-predicted using the base imagery pair in front of them. Thus, DOY 102 and DOY 122 were underestimated with wheat growing, while DOY 137 was overestimated with wheat aging. DOY 87 was backward-predicted using the base imagery pair on DOY 102, so it was overestimated. Perhaps because of the different prediction direction of DOY 87, the variation was slightly larger than that of other dates.

The four existing STF methods tested here performed poorly for the boundary predictions. The new STF framework greatly improved boundary predictions, but there was still a significant gap between heterogeneous regions and homogeneous regions. Delineating boundary changes is extremely difficult, because the boundaries are not characterizable in any of the coarse pixels, and the information on boundary changes is usually not captured by the available fine pixels. However, it is observed that EVI2 presents less variation than NDVI and is more stable. The calculation of EVI2 introduces the parameters for background adjustment and atmospheric correction on the basis of NDVI, which can reduce the influence of atmospheric and soil noise and make it more robust. Our results suggest that the STF framework developed in this research has potential for blending UAV and satellite imagery, and is beneficial for monitoring phenological change during crop growth intensively and at low cost. However, it is still challenging to predict the shape or boundary changes of objects. In addition, this study only verified the variation in VIs of the synthetic imagery during one growth cycle. The exploration of harmonizing UAV and satellite imagery for quantitative inversion will be focused on in the future, and the verification of more growth cycles is needed.

## 6. Conclusions

This research presents an attempt at STF of UAV and satellite imagery. Considering the difference between UAV and satellite platforms, an STF framework for UAV and satellite imagery is proposed based on the existing STF methods, and its application potential for growth monitoring is further explored. The qualitative and quantitative results show that four tested methods covering spatial unmixing, spatial weighting, and hybrid methods fail to provide accurate predictions. STARFM and FSDAF perform better than UBDF and Fit-FC, but the spatial details are still poorly predicted. Considering the runtime and accuracy, STARFM was identified as more suitable than the other three methods to combine with CACAO, producing CA-STARFM. CA-STARFM significantly improves the spectral and spatial fusion effect at daily temporal resolution, especially for object boundary and spatial texture, with the absolute values of EDGE and LBP decreasing by a maximum of

more than 0.25 and 0.10, respectively. Further, the STF framework is competent to describe the seasonal dynamics of winter wheat, the synthesized VIs are consistent with the UAV observations, and the outliers are mainly distributed at the boundary of objects. Therefore, the delineating of boundary changes should be focused on further. This study suggests that the STF of UAV and satellite imagery is promising. The proposed STF framework has excellent potential for harmonizing UAV and satellite imagery to monitor crop growth intensively and in greater detail.

**Author Contributions:** Conceptualization, Y.L. and S.T.; methodology, Y.L. and S.T.; software, Y.L.; validation, Y.L., W.Y. and W.W.; formal analysis, Y.L.; investigation, Y.L. and S.A.; resources, S.T. and W.W.; data curation, Y.L.; writing—original draft preparation, Y.L.; writing—review and editing, W.Y., S.A. and W.W.; visualization, Y.L.; supervision, W.G.; project administration, J.J.; funding acquisition, Y.L. and S.T. All authors have read and agreed to the published version of the manuscript.

**Funding:** This research was funded by the Natural Science Foundation of Hebei Province of China (No. D2022407001) and the Soft Science Project of Hebei Science and Technology Program (22557672D and 22557674D).

**Data Availability Statement:** Not applicable.

**Acknowledgments:** Thanks to the authors for publicly providing the codes used in this study.

**Conflicts of Interest:** The authors declare no conflict of interest.

## Appendix A

**Table A1.** Code and software settings.

| Procedure | Content | Method | Number of Classes | Moving Window Size | Number of Similar Pixels | Software |
|---|---|---|---|---|---|---|
| imagery consistency processing | geo-registration | | | shift by 0.5 m, maximum 2 m. | | Matlab |
| | radiometric normalization | | | 2 × 2 S2-MSI pixels moving window | | Matlab |
| preliminary predictions using existing STF methods | | STARFM | 5 | 31 × 31 UAV pixels | N/A | IDL |
| | | UBDF | 5 | 9 × 9 S2-MSI pixels | N/A | IDL |
| | | Fit-FC | N/A | 2 × 2 S2-MSI pixels in RM / 31 × 31 UAV pixels in SF and RC | 20 | Matlab |
| | | FSDAF | 5 | 31 × 31 UAV pixels | 20 | IDL |
| reconstruction with CACAO | | CACAO | | 2-degree polynomial and 17-day temporal window in S-G filtering; 3 UAV imagery and a maximum shift of 10 days in model fitting. | | Matlab |
| accuracy assessment | framework proposed by Zhu [39]. | | | N/A | | IDL |

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
