# Peer review of "A Spatio-Temporal Fusion Framework of UAV and Satellite Imagery for Winter Wheat Growth Monitoring"

_drones, doi:10.3390/drones7010023_

Round 1
Reviewer 1 Report
Using drone and satellite imagery fusion to monitor winter wheat growth is interesting but not novel. The advantages of this article should be clearly stated after comparison with previous relevant literature (suggested list description). At the same time, there are some suggestions for reference.
1. How to overlap or fuse the S2-MSI image and UAV image.
2. Winter wheat growth is cyclical, and the article only has one cycle (one case), which is not representative. It is recommended to add explanations or raise limitations in conclusion.
3. The complete UAV setting method and equipment information should be proposed.
4. In the overlapping and correction of images, is the area of the winter wheat field used as the sampling frame or the feature points as the sampling and correction points? This is critical to the correctness of the overlapping items.
5. UAV will affect image pixels at different heights and speeds. It is recommended to explain the research design. References:
The use of UAV to detect solar module fault conditions of a solar power farm with IR and visual image analysis
Using drones for thermal imaging photography and building 3-D images to analyze the defects of solar modules
Reviewer 2 Report
The manuscript is enough well written and structured. An interesting topic is depicted anyway before going ahead some revision have to be performed. If authors will follow the advice given I will certainly underline this article to be published on Drones.
Firstly, I suggest you to better point out in the introduction section the main aim and purposes of your study. At this time is not enough well clear. Therefore I suggest you to decline in the last part of the introduction a clear sentence in which you describe the aim and proposes of your study.
Moreover, I suggest to consider to include in the introduction the role of a join use of satellite and drones which is becoming very popular. You do a nice introduction except the aim part but i deeply suggest to include this to perform a 360° state of art and improve the quality of your work. Thus, to help you in doing this I advice you to include these manuscript on your work:
https://doi.org/10.3390/rs12213542
https://doi.org/10.1016/j.scitotenv.2019.01.112
Then I suggest you to include for each maps in figure the EPSG or the Datum, Reference System, Nominal and Representation scale. This is very important in a Drones journal that deal with cartography.
Moreover, I suggest you to include in material and methods the drone parameters that are crucial like flight speed, altitude acqusition, focal caractheristics, GSD, position of the camera respect to the ground nadir or which angle? Consider also to include calibration curve of the sensors how do you perform it? And what is it uncertainity. Describe the drones and which software you perform and settings to process. This will help other reseachers in terms of scalability and improve the quality and scientific robustness of your work!
Please consider also to describe how you orthorectify the images you have an RTK drones or you put target (in this case consider to include and image with them on the ground)
Considering this last point please include a table with errors RMSE
Finally consider to include statistics code or s software settings in a supplementary material.
Results is fine, consider only to better describe the graphs
Conclusion is well written.
Round 2
Reviewer 1 Report
The authors have improved their manuscript according to the reviewers' comments. It can be accepted for publication in this version.
Reviewer 2 Report
The authors well aimed the suggestion given therefore the manuscript can be published.